# Sensor Fault Diagnosis Using a Machine Fuzzy Lyapunov-Based Computed Ratio Algorithm

**DOI:** 10.3390/s22082974

**Published:** 2022-04-13

**Authors:** Shahnaz TayebiHaghighi, Insoo Koo

**Affiliations:** Department of Electrical, Electronic and Computer Engineering, University of Ulsan, Ulsan 680-749, Korea; nazitayebi5@liveuou.kr

**Keywords:** internal combustion engine, sensor anomaly detection, Gaussian autoregressive method, fuzzy approach, computed ratio observer, Lyapunov robust method, support vector machine

## Abstract

Anomaly identification for internal combustion engine (ICE) sensors has become an important research area in recent years. In this work, a proposed indirect fuzzy Lyapunov-based computed ratio observer integrated with a support vector machine (SVM) was designed for sensor fault classification. The proposed fuzzy Lyapunov-based computed ratio observer integrated with SVM has three main layers. In the preprocessing (first) layer, the resampled root mean square (RMS) signals are extracted from the original signals to the designed indirect observer. The second (observation) layer is the principal part with the proposed indirect fuzzy sensor-fault-classification technique. This layer has two sub-layers: signal modeling and estimation. The Gaussian autoregressive-Laguerre approach integrated with the fuzzy approach is designed for resampled RMS fuel-to-air-ratio normal signal modeling, while the subsequent sub-layer is used for resampled RMS fuel-to-air-ratio signal estimation using the proposed fuzzy Lyapunov-based computed ratio observer. The third layer, for residual signal generation and classification, is used to identify ICE sensor anomalies, where residual signals are generated by the difference between the original and estimated resampled RMS fuel-to-air-ratio signals. Moreover, SVM is suggested for residual signal classification. To test the effectiveness of the proposed method, the results are compared with two approaches: a Lyapunov-based computed ratio observer and a computed ratio observer. The results show that the accuracy of sensor anomaly classification by the proposed fuzzy Lyapunov-based computed ratio observer is 98.17%. Furthermore, the proposed scheme improves the accuracy of sensor fault classification by 8.37%, 2.17%, 6.17%, 4.57%, and 5.37% compared to other existing methods such as the computed ratio observer, the Lyapunov-based computed ratio observer, fuzzy feedback linearization observation, self-tuning fuzzy robust multi-integral observer, and Kalman filter technique, respectively.

## 1. Introduction

In recent decades, there has been considerable growth in the evolution of smart systems in processing machines. The primary centers of attention (accuracy and reliability) in this industry are highly dependent on data from various sensors. Many modernized processes in manufacturing are equipped with sensors to collect process-related data to find anomalies occurring in the processes as well as for observing them. Because sensors have become one of the principal parts, accuracy and efficiency in sensor operations are more significant issues. A sensor is a component that monitors the output environment while the controller adjusts its behavior according to these sensor readings. Speed sensors, steering angle sensors, gyroscopes, and acceleration sensors have been installed to measure a machine’s functional state and to use the gathered information for controlling the system. If faults happen in these sensors or actuators, the machine may fail to conduct the accurate operations needed for control, which may lead to a bad situation [1]. This means that sensors may fail, or they might be affected by other devices, or take faulty actions [2]. Furthermore, sensors are prone to malfunctions, failure, and rapid attrition. These conditions cause sensors to provide inaccurate and/or erroneous data. In general, faults in machine sensors lead to intense safety issues because these sensors are embedded to detect automatic functions in the machines. Hence, the immediate identification of sensor faults is a serious requirement in order to take corrective action to decrease the influence of the fault. We can infer that sensors send measured data to the controller, and the controller acts based on its programming; thus, the main objective is to prevent or reduce failure of system control caused by a sensor fault [3]. The significance of fault detection can be generalized to all parts of a machine, including combustion engines and fuel systems. Normally, combustion engines are divided into two groups: external combustion engines and internal combustion engines. The internal combustion engine (ICE) is used in industries as the main power source in different applications, such as gas compressors, generating sets, air compressors, and vehicles. These engines convert energy. Indeed, these devices in alternators and compressors convert the chemical energy of the fuel into mechanical rotational energy. In ICEs, a proper ratio of air and fuel is needed for proper combustion. One specific controller is utilized for the air–fuel ratio (AFR) control in these engines. If the AFR reading is not correct, combustion will not be efficient. This fuel control system generally acts on feedback from sensors applied in different locations. The presence of faults in sensors means accuracy and reliability in controller operations—and of course, operations of the fuel system itself—are endangered. Therefore, it is extremely desirable to make this system reliable to avoid failure of the entire machine and/or major breakdowns due to sensor faults. Motivated by the discussion above, sensor fault diagnosis in internal combustion engines has been considered by many researchers [4].

To prevent major breakdowns from sensor faults, different methods of fault detection identification (FDI) are garnering attention among researchers. Toward this objective, different ideas and methods for FDI techniques, including data-driven techniques, model-reference methods, and hybrid algorithms, have been presented [3,5,6,7]. Based on the information provided, different methods are applied to the issue of sensor fault detection and diagnosis based on model information [8]. Control-based methods using model information [9] are utilized for sensor fault detection and diagnosis; moreover, different model-reference methods, such as observation techniques, can be employed for fault diagnosis. The evaluation and efficiency of model-based techniques invariably rely on the model of the system employed. One of the principal techniques in the model-reference approach is the observation technique. Different categories of observation techniques have been utilized and can be defined in two main groups: linear observation and nonlinear observation. The linear observation techniques include the proportional integral observer [10], the proportional multi-integral observer [11], and the proportional integral derivative observer [12]. These techniques lack robustness, reliability, and nonlinear signal estimation, although a positive aspect is the simplicity of implementation. To overcome the limitations, nonlinear observation approaches have been recommended. Piltan and Kim improved the performance of the linear observation technique with a nonlinear-based approach that is a feedback linearization technique [13]. In general, nonlinear observation methods are divided into three groups: (a) classic observation methods, such as feedback linearization, sliding-mode, Lyapunov, and backstepping approaches [14,15,16,17]; (b) intelligent observation methods, such as neural networks and fuzzy logic observers [18,19]; and (c) hybrid methods, where a combination of classic and intelligent methods is used [20]. Therefore, to overcome complexity and operation under uncertain conditions related to model-reference techniques, it is preferable to use them in combination with an artificial intelligence approach and a data-driven technique. Given that data-driven and artificial-intelligence-based methods also have limitations in reliability and robustness, the hybrid approach is expected to provide an effective result in fault diagnosis issues. In this work, the proposed hybrid observer is recommended for sensor fault identification.

Different techniques for signal/system modeling have been introduced in various articles, which can be categorized into two groups: mathematical-based modeling and data-based modeling. To model nonlinear and uncertain systems, data-based modeling algorithms were suggested [21]. This technique can be divided into two principal schemes: artificial intelligence and the identification approach. The application of artificial intelligence modeling, such as neural-network and fuzzy algorithms, was utilized in [22]. Furthermore, identification techniques in system modeling including autoregressive, autoregressive with external input, autoregressive-Laguerre, nonlinear autoregressive, and Gaussian algorithms were introduced in [23]. In general terms, modeling systems with Laguerre filters has been intensively studied over the years, because using this kind of filter ends up reducing modeling parameter complexity, and its simple structure is the result of a linear combination of Laguerre filters. This technique was proposed and started by Arnold [24] and was utilized in system identification [25,26] and control [27,28]. In this research article, a nonlinear-based system identification technique is proposed for signal modeling.

To classify the state of signals, an effective technique for decision making is needed. Decision-making algorithms fall into two main groups: classic algorithms and machine learning-based algorithms. Classic algorithms, such as rule-based methods [29], have specifications such as rapid classification for new instances and offer easy interpretation. Machine-learning-based algorithms such as decision trees [30] and the support vector machine (SVM) [31] have been used by many researchers in recent years. The support vector machine is an accurate and robust method for both linear and nonlinear conditions. SVM generalization efficiency either matches or is remarkably better than that of competing techniques [32]. In this research, SVM is used for signal classification.

In this study, one fault-tolerant fuel control system for the IC gas engine model available in MATLAB is introduced. In this model, four different sensors are used for an air-fuel ratio in the control system of a gas engine. These sensors include the throttle position sensor (TH) to feed back the air throttle physical status to the engine controller and the manifold air pressure (MAP) sensor for the calculation of air density and air mass flow rates. In fact, the MAP sensor supplies suction pressure of the airflow at the input manifold of the engine. The engine speed (SP) sensor provides feedback on engine speed to the controller, and the exhaust gas oxygen (EGO) sensor detects the quantity of oxygen in the exhaust gas of the engine for adjusting the fuel supply [3,33].

In this paper, ICE motor sensor fault diagnosis using the proposed hybrid approach is conducted. The proposed hybrid method for ICE sensor fault diagnosis has three stages. In the first stage, ICE sensor signals are resampled, and a feature is extracted from the resampled signals. After that, the proposed hybrid approach based on a fuzzy Lyapunov-based computed ratio algorithm provides signal estimation. Before signal estimation, the proposed signal modeling approach using the combination of Gaussian regression, Laguerre, and fuzzy techniques to overcome mathematical modeling problems are recommended. Now, the state-space signal extracted by the proposed signal modeling approach is used in the proposed observer to improve the power of signal estimation. The proposed observer integrates the computed ratio observer, the Lyapunov technique, and the fuzzy approach. In the last step, SVM is proposed for sensor fault detection and identification. This research makes the following contributions:ICE sensor signal modeling based on the proposed modeling algorithm is a combination of Gaussian regression, Laguerre, and fuzzy techniques.The estimation of motor sensor signals by the proposed computer ratio observer will be more robust and adaptable in combination with the Lyapunov technique and the fuzzy approach.

This article has the following sections. The proposed scheme is introduced in Section 2. In it, preprocessing is introduced first. After that, the motor sensor signal is modeled using the fuzzy Gaussian autoregressive-Laguerre approach. Next, the motor sensor signals are estimated using a fuzzy Lyapunov-based computed ratio observer. Finally, these signals are classified using SVM. In Section 3, the proposed scheme is tested, and the results are discussed. The conclusions are presented, and future work is examined in the final section.

## 2. The Proposed Scheme

For fault diagnosis of internal combustion engine sensors, this scheme has the following steps: (1) sensor signal preprocessing (resampling and featuring), (2) sensor signal estimation, and (3) sensor signal classification. Figure 1 shows a block diagram of the proposed scheme for ICE sensor fault diagnosis.

The preprocessing unit is used to resample normal and abnormal ICE sensor data and extract the root mean square (RMS) feature from them. The resampled RMS signals from ICE sensor estimation have two important sub-parts: a resampled RMS signal of a normal condition, and a resampled RMS signal using the proposed observer of normal and abnormal conditions. To model the resampled RMS signal of an ICE sensor, first, the Gaussian autoregressive technique is suggested. To improve modeling robustness, the combination of a Gaussian autoregressive technique and a Laguerre filter is recommended. Moreover, to reduce errors in sensor modeling of a healthy condition, the fuzzy algorithm is applied to the Gaussian autoregressive technique and Laguerre combination, extracting the state-space equation. After modeling the resampled RMS signal of an ICE sensor in a healthy condition, the proposed observation algorithm is recommended for signal estimation. Thus, first, the fuzzy Gaussian autoregressive-Laguerre combination and the computed ratio observer are suggested for ICE sensor signal estimation under unknown conditions. After that, the Lyapunov technique is applied to the fuzzy Gaussian autoregressive-Laguerre combination and the computed ratio observer to improve the robustness of the estimation algorithm. Next, to improve the accuracy of the proposed observer, the fuzzy logic approach is applied to the fuzzy Gaussian autoregressive-Laguerre technique and Lyapunov-based computed ratio observer combination. To classify the state of an ICE sensor fault, a machine learning approach with an SVM is recommended.

### 2.1. Preprocessing

For an indirect observer, the first step is preprocessing. In this, normal and abnormal ICE sensor signals are resampled. After that, based on the following definition, the RMS of the resampled ICE sensor signals is represented as follows.
(1)$$S_{rms}=\sqrt{\frac{1}{M}\overset{M}{\underset{j=1}{\sum }}S^{2}}$$

Here, $S_{rms},\;\;M$, and S denote the resampled RMS ICE sensor value for the window, the number of windows, and the original ICE sensor signal in the window, respectively.

### 2.2. RMS Resample Modeling of Healthy ICE Sensor Signals

Based on Figure 1, the RMS resampled normal ICE sensor signal is first modeled using the Gaussian autoregressive (GA) algorithm. The formulation of the GA algorithm for modeling the sensor signal is represented as follows.
(2)$$\{\begin{array}{l} S_{x-GA}\left(k+1\right)=\left[\chi_{\mathrm{GA}}S_{x-GA}\left(k\right)+\chi_{i}S_{i}\left(k\right)\right]+e_{GA}\left(k\right)+\varphi_{GA}\left(k\right)\\ S_{o-GA}\left(k\right)={\left(\chi_{o-GA}\right)}^{T}\left(\chi_{n}\right)\chi_{\mathrm{GA}}{}^{-1}S_{x-GA}\left(k\right) \end{array}$$

Here, $S_{x-GA}\left(k\right),\;\chi_{\mathrm{GA}},\;S_{i}\left(k\right),\;e_{GA}\left(k\right),\;\varphi_{GA}\left(k\right),\;S_{o-GA}\left(k\right),\;$ and $\left(\chi_{i},\;\chi_{o-GA},\chi_{n}\right)$ are the state of the ICE sensor signal in a healthy condition using the GA technique, the covariance matrix for the ICE sensor signal in a healthy condition using the GA technique, the input of the ICE sensor signal, the error in the signal from ICE sensor modeling using the GA approach, the uncertainty in the signal from ICE sensor modeling under normal conditions using the GA approach, the state output for the signal from ICE sensor modeling using the GA approach, and the coefficients of the GA approach, respectively. Moreover, the covariance matrix ($\chi_{\mathrm{GA}}$) is represented in the following definition:(3)$$\chi_{\mathrm{GA}}=\chi_{x}{}^{2}e^{\left(-0.5S_{x-GA}{}^{T}{∁}^{-1}S_{x-GA}\right)}+\partial \;\&\;∁=diag\left(Y^{2}\right)$$
where $\chi_{x}$ is variance of the signal from ICE sensor modeling in a normal condition, ∂ is the noise variance, and Y is the width of the kernel. Moreover, the error for the ICE sensor modeling signal using the GA approach, $e_{GA}\left(k\right)$, can be represented as follows.
(4)$$e_{GA}\left(k\right)=S_{o-GA}\left(k\right)-S_{o-GA}\left(k-1\right)$$

Based on Equation (2), the uncertainty of the signal from ICE sensor modeling under normal conditions using the GA approach, ($\varphi_{GA}\left(k\right)$), is one of the critical challenges in signal modeling. To improve the robustness and reduce the effect of uncertainties, the combination of the GA approach and the Laguerre filter (GAL) is recommended. Thus, the state-space equation of the GAL algorithm is represented as follows.
(5)$$\{\begin{array}{l} S_{x-GAL}\left(k+1\right)=\left[\chi_{\mathrm{GAL}}S_{x-GAL}\left(k\right)+\chi_{i}S_{i}\left(k\right)+\chi_{s}S_{o-GAL}\left(k\right)\right]+e_{GAL}\left(k\right)\\ +\varphi_{GAL}\left(k\right)\\ S_{o-GAL}\left(k\right)={\left(\chi_{o-GAL}\right)}^{T}\left(\chi_{n}\right)\chi_{\mathrm{GAL}}{}^{-1}S_{x-GAL}\left(k\right) \end{array}$$

Here, $S_{x-GAL}\left(k\right),\;\chi_{\mathrm{GAL}},\;e_{GAL}\left(k\right),\;\varphi_{GAL}\left(k\right),\;S_{o-GAL}\left(k\right),$ and $\left(\chi_{s}\right)$ are the state of the ICE sensor signal in a healthy condition using the combination of the GAL technique, the covariance matrix for the ICE sensor signal in a healthy condition using the GAL technique, the error for the ICE sensor modeling signal using the GAL approach, the uncertainty of the ICE sensor modeling signal in a normal condition using the GAL technique, the state output for the ICE sensor modeling signal using the GAL approach, and the coefficients of the GAL approach, respectively. Furthermore, the error for ICE sensor signal modeling using the GAL approach combination, $e_{GAL}\left(k\right)$, can be represented as follows.
(6)$$e_{GAL}\left(k\right)=S_{o-GAL}\left(k\right)-S_{o-GAL}\left(k-1\right)$$

Additionally, regarding Equations (5) and (6), the error for the ICE sensor-modeling signal is an important factor for system/signal modeling. To reduce the error in the ICE sensor-modeling signal, the combination of a fuzzy algorithm and the GAL technique, called FGAL, is recommended. The fuzzy algorithm is a type of artificial intelligence technique that can be used in system modeling and control. Thus, the state-space equation for the FGAL algorithm is as follows.
(7)$$\{\begin{array}{l} S_{x-FGAL}\left(k+1\right)=\left[\chi_{\mathrm{FGAL}}S_{x-FGAL}\left(k\right)+\chi_{i}S_{i}\left(k\right)+\chi_{s}S_{o-FGAL}\left(k\right)\right]\\ +e_{FGAL}\left(k\right)+\chi_{f}S_{f}\left(k\right)+\varphi_{FGAL}\\ S_{o-FGAL}\left(k\right)={\left(\chi_{o-FGAL}\right)}^{T}\left(\chi_{n}\right)\chi_{\mathrm{FGAL}}{}^{-1}S_{x-FGAL}\left(k\right) \end{array}$$

In addition, the error in the ICE sensor modeling signal using the FGAL approach, $e_{FGAL}\left(k\right)$, can be introduced using the following definition:(8)$$e_{FGAL}\left(k\right)=S_{o-FGAL}\left(k\right)-S_{o-FGAL}\left(k-1\right)$$

Here, $S_{x-FGAL}\left(k\right),\;\chi_{\mathrm{FGAL}},\;e_{FGAL}\left(k\right),\;\varphi_{FGAL}\left(k\right),\;S_{o-FGAL}\left(k\right),\;S_{f}\left(k\right),$ and $\left(\chi_{f}\right)$ are the state of the ICE sensor signal in a healthy condition using FGAL, the covariance matrix for the ICE sensor signal in a healthy condition using FGAL, the error in the ICE sensor modeling signal using FGAL, the uncertainty in the ICE sensor modeling signal in a normal condition using FGAL, the state output for the ICE sensor modeling signal using FGAL, the normal ICE sensor modeling signal using the fuzzy approach, and the coefficients of FGAL, respectively. Furthermore, a normal signal from ICE sensor modeling using the fuzzy approach, $S_{f}\left(k\right)$, can be represented as follows:(9)$$S_{f}\left(k+1\right)=\frac{\sum^{\;}S_{f}\left(k\right)\prod^{\;}\mu \left(e_{GAL}\left(k\right)\right)}{\sum^{\;}\prod^{\;}\mu \left(e_{GAL}\left(k\right)\right)}$$
where $\mu \left(e_{GAL}\left(k\right)\right)$ is the membership of the error in the fuzzy approach.

### 2.3. RMS Resample Estimation of ICE Sensor Signals in Unknown Conditions

Based on Figure 1 and Equation (7), the resampled ICE sensor signal is modeled in a healthy condition, and the state-space equations are extracted from the signal. To evaluate the power of signal categorization, this section introduces the observation technique. The observer is used to estimate the signal based on the proposed modeling approach. First, the computed ratio observer estimates the nonlinear ICE sensor signals in unknown conditions. Thus, the state-space equation of the fuzzy Gaussian autoregressive-Laguerre computed ratio (FGAL-CR) algorithm is represented as follows.
(10)$$\left\{ \begin{array}{l} S_{x-FGAL-CR}\left(k+1\right)=\left[\chi_{\mathrm{FGAL}}S_{x-FGAL-CR}\left(k\right)+\chi_{i}S_{i}\left(k\right)+\chi_{s}S_{o-FGAL-CR}\left(k\right)\right]\\ +e_{FGAL}\left(k\right)+\chi_{f}S_{f}\left(k\right)+\varphi_{FGAL-CR}+\chi_{CR}\left(S_{rms}\left(k\right)-S_{o-FGAL-CR}\left(k\right)\right)\\ S_{o-FGAL-CR}\left(k\right)={\left(\chi_{o-FGAL-CR}\right)}^{T}\left(\chi_{n}\right)\chi_{\mathrm{FGAL}}{}^{-1}S_{x-FGAL-CR}\left(k\right) \end{array}\right.$$

The uncertainty of the ICE sensor estimation signal in unknown conditions using the fuzzy Gaussian autoregressive-Laguerre computed ratio algorithm ($\varphi_{FGAL-CR}$) can be introduced using the following definition.
(11)$$\begin{array}{ll} \varphi_{FGAL-CR}\left(k+1\right)= & \varphi_{FGAL-CR}\left(k\right)+\chi_{CR}\left(S_{rms}\left(k\right)-S_{o-FGAL-CR}\left(k\right)\right)\\ & +\chi_{CR}\times sgn||S_{rms}\left(k\right)-S_{o-FGAL-CR}\left(k\right)|| \end{array}$$

Here, $S_{x-FGAL-CR}\left(k\right),\varphi_{FGAL-CR}\left(k\right),\;S_{o-FGAL-CR}\left(k\right),S_{rms}\left(k\right),$ and $\left(\chi_{CR}\right)$ are the state estimation of the ICE sensor signal in an unknown condition using the FGAL-CR approach, the uncertainty of signal estimation for the ICE sensor in an unknown condition using the FGAL-CR approach, the output of state estimation for the signal of the ICE sensor using the FGAL-CR approach, the original signal of the ICE sensor, and the coefficients of the FGAL-CR approach, respectively. The foremost significant challenge of the computed ratio observer is robustness, especially because of uncertainty estimation and variations in motor speed. To address these issues, the application of the Lyapunov algorithm in the FGAL-CR approach is proposed here, called the fuzzy Gaussian autoregressive-Laguerre Lyapunov-based computed ratio approach (FGAL-LCR). Hence, the state-space definition is represented as follows.
(12)$$\left\{ \begin{array}{l} S_{x-FGAL-LCR}\left(k+1\right)=\left[\chi_{\mathrm{FGAL}}S_{x-FGAL-LCR}\left(k\right)+\chi_{i}S_{i}\left(k\right)+\chi_{s}S_{o-FGAL-LCR}\left(k\right)\right]\\ +e_{FGAL}\left(k\right)+\chi_{f}S_{f}\left(k\right)+\varphi_{FGAL-LCR}+\chi_{CR}\left(S_{rms}\left(k\right)-S_{o-FGAL-LCR}\left(k\right)\right)\\ S_{o-FGAL-LCR}\left(k\right)={\left(\chi_{o-FGAL-LCR}\right)}^{T}\left(\chi_{n}\right)\chi_{\mathrm{FGAL}}{}^{-1}S_{x-FGAL-LCR}\left(k\right) \end{array}\right.$$

To improve robustness, the Lyapunov nonlinear function is applied to the uncertainty of ICE sensor signal estimation in unknown conditions. Thus, the uncertainty of ICE sensor signal estimation in unknown conditions using the FGAL-LCR algorithm ($\varphi_{FGAL-LCR}$) is represented as follows.
(13)$$\begin{array}{ll} \varphi_{FGAL-LCR}\left(k+1\right) & =\varphi_{FGAL-LCR}\left(k\right)+\chi_{CR}\left(S_{rms}\left(k\right)-S_{o-FGAL-LCR}\left(k\right)\right)\\ & +\chi_{CR}\times sgn||S_{rms}\left(k\right)-S_{o-FGAL-LCR}(k)||\\ & +\vartheta_{\gamma }\left(e_{FGAL}\left(k\right),S_{rms}\left(k\right),\varphi_{FGAL-LCR}\right) \end{array}$$

Here, $S_{x-FGAL-LCR}\left(k\right),\varphi_{FGAL-LCR}\left(k\right),\;S_{o-FGAL-LCR}\left(k\right),$ and $\left(\vartheta_{\gamma }\left(e_{FGAL}\left(k\right),S_{rms}\left(k\right),\varphi_{FGAL-LCR}\right)\right)$ are the state estimations of the ICE sensor signal in unknown conditions using the FGAL-LCR approach, the uncertainty of the signal estimation for the ICE sensor in unknown conditions using the FGAL-LCR approach, the output of state estimation for the ICE sensor signal using the FGAL-LCR approach, and the nonlinear Lyapunov function, respectively. Moreover, the nonlinear Lyapunov function, $\vartheta_{\gamma }\left(e_{FGAL}\left(k\right),S_{rms}\left(k\right),\varphi_{FGAL-LCR}\right)$, is a Hamilton–Jacobi function and a differential of uncertainty function, represented by the following:(14)$$\vartheta_{\gamma }\left(e_{FGAL}\left(k\right),S_{rms}\left(k\right),\varphi_{FGAL-LCR}\right)={ℋ}_{\gamma }\left(e_{FGAL}\left(k\right),S_{rms}\left(k\right)\right)+{ℒ}_{\gamma }\left(e_{FGAL}\left(k\right)\right)\varphi_{FGAL-LCR}$$
where ${ℋ}_{\gamma }\left(e_{FGAL}\left(k\right),S_{rms}\left(k\right)\right)$, and ${ℒ}_{\gamma }\left(e_{FGAL}\left(k\right)\right)\varphi_{FGAL-LCR}$ are the Hamilton–Jacobi observation function and the differentiated uncertainty estimation function, respectively. To reduce the difference between an original unknown signal and an estimated unknown signal and to reduce the effect of uncertainties, the fuzzy logic approach is suggested for this step. Thus, the application of the fuzzy algorithm in the FGAL-LCR approach here is called the fuzzy Gaussian autoregressive-Laguerre fuzzy Lyapunov-based computed ratio (FGAL-FLCR) approach. Hence, the state-space definition is introduced by the following equation.
(15)$$\left\{ \begin{array}{l} S_{x-FGAL-FLCR}\left(k+1\right)=[\chi_{\mathrm{FGAL}}S_{x-FGAL-FLCR}\left(k\right)+\chi_{i}S_{i}\left(k\right)+\\ \;\;+\chi_{s}S_{o-FGAL-FLCR}\left(k\right)]+e_{FGAL}\left(k\right)+\chi_{f}S_{f}\left(k\right)+\varphi_{FGAL-FLCR}\\ \;\;\;\;\;\;\;+\chi_{CR}\left(S_{rms}\left(k\right)-S_{o-FGAL-FLCR}\left(k\right)\right)\\ S_{o-FGAL-FLCR}\left(k\right)={\left(\chi_{o-FGAL-FLCR}\right)}^{T}\left(\chi_{n}\right)\chi_{\mathrm{FGAL}}{}^{-1}S_{x-FGAL-FLCR}\left(k\right) \end{array}\right.$$

To improve accuracy, the fuzzy function is applied to the uncertainty of ICE sensor signal estimation in unknown conditions. Thus, the uncertainty of ICE sensor signal estimation in unknown conditions using the proposed FGAL-FLCR algorithm ($\varphi_{FGAL-FLCR}$) is represented as follows.
(16)$$\begin{array}{ll} \varphi_{FGAL-FLCR}\left(k+1\right) & =\varphi_{FGAL-FLCR}\left(k\right)+\chi_{CR}\left(S_{rms}\left(k\right)-S_{o-FGAL-FLCR}\left(k\right)\right)\\ & +\chi_{CR}\times sgn||S_{rms}\left(k\right)-S_{o-FGAL-FLCR}(k)||\\ & +\vartheta_{\gamma }\left(e_{FGAL}\left(k\right),S_{rms}\left(k\right),\varphi_{FGAL-FLCR}\right)+\chi_{f}S_{f}\left(k\right) \end{array}$$

Here, $S_{x-FGAL-FLCR}\left(k\right),\varphi_{FGAL-FLCR}\left(k\right),$ and $\left(S_{o-FGAL-FLCR}\left(k\right)\right)$ are the state estimation of the ICE sensor signal in an unknown condition using the proposed FGAL-LCR approach, the uncertainty of signal estimation for the ICE sensor in an unknown condition using the proposed FGAL-LCR approach, and the output of state estimation for the ICE sensor signal using the proposed FGAL-LCR approach, respectively. Based on Equations (10), (12), and (15), three observation techniques are introduced to estimate the RMS resampled ICE sensor signals: the fuzzy Gaussian autoregressive-Laguerre computed ratio technique, the fuzzy Gaussian autoregressive-Laguerre Lyapunov-based computed ratio method, and the proposed fuzzy Gaussian autoregressive-Laguerre fuzzy Lyapunov-based computed ratio approach, respectively.

### 2.4. Generate the RMS Resampled Residual of ICE Sensor Signals and the Fault Decision

Based on Figure 1, in this section, the residual RMS resampled ICE sensor signals based on the FGAL-CR technique, the FGAL-LCR method, and the proposed FGAL-FLCR approach are generated. Thus, the residual of the RMS resampled ICE sensor signal based on the FGAL-CR technique, $Res_{FGAL-CR}\left(k\right)$, is generated as follows.
(17)$$Res_{FGAL-CR}\left(k\right)=\left(S_{rms}\left(k\right)-S_{o-FGAL-CR}\left(k\right)\right)$$

Moreover, the residual of the RMS resampled ICE sensor signal based on the FGAL-LCR technique, $Res_{FGAL-LCR}\left(k\right)$, is generated using the following definition.
(18)$$Res_{FGAL-LCR}\left(k\right)=\left(S_{rms}\left(k\right)-S_{o-FGAL-LCR}\left(k\right)\right)$$

Additionally, the residual of the RMS resampled ICE sensor signal based on the proposed FGAL-FLCR technique, $Res_{FGAL-FLCR}\left(k\right)$, is generated using the following definition:(19)$$Res_{FGAL-FLCR}\left(k\right)=\left(S_{rms}\left(k\right)-S_{o-FGAL-FLCR}\left(k\right)\right)$$

To classify the RMS resampled residual of an ICE sensor’s signal, the support vector machine is recommended. Algorithm 1 shows the steps to designing the proposed machine fuzzy Lyapunov-based computed ratio algorithm for ICE sensor fault diagnosis.
**Algorithm 1:** Proposed machine fuzzy Lyapunov-based computed ratio algorithm for ICE sensor anomaly diagnosis.**1. Preprocessing**1. Compute the RMS feature of the ICE sensor resample signals: Equation (1) **2.1. Resampled RMS feature modeling for ICE sensor signals**2.1.1. Resample RMS feature modeling for ICE sensor signals in healthy conditions using Gaussian autoregressive (GA) algorithm: Equations (2) and (3) 2.1.2. Improve robustness of the GA method using Laguerre filter approach, namely GAL: Equation (5)2.1.3. Reduce the error in resampled RMS feature modeling using application of fuzzy modeling in the GAL algorithm to design the proposed FGAL feature modeling approach: Equation (7)**2.2. Resampled RMS feature estimating of ICE sensor signals**2.2.1. Evaluate the power of signal categorization using computed ratio observer (FGAL-CR): Equations (10) and (11)2.2.2. Improve robustness of the FGAL-CR method using Lyapunov approach (FGAL-LCR): Equations (12) and (13)2.2.3. Improve power of uncertain resampled RMS feature estimation using application of fuzzy estimating to design the proposed FGAL-FLCR: Equations (15) and (16)**3. Residual resampled RMS feature generation and signal classification**3.1. Generate resampled RMS residual features using the difference between original resampled RMS feature and estimated RMS feature: Equation (19)3.2. Sensor fault diagnosis: use SVM [32]

## 3. Data, Results, and Discussion

In [3,33,34], the fuel control and modeling technique in MATLAB/Simulink for ICE is reported with a complete description about fault diagnosis and control. The solver option in MATLAB/Simulink is variable-step with ode45 (Dormand-Prince) and the tolerance is set to 1 × 10^−6^. In this research, faults in four sensors—the throttle sensor, the speed sensor, the exhaust gas oxygen (EGO) sensor, and the manifold absolute pressure (MAP) sensor—are simulated using MATLAB [3,33]. Based on [34], the analytical redundancy for sensors using the look-up tables has been used for fault simulation in sensors. This model is also used for the reliability enhancement of the AFR control system. Furthermore, the air density and air mass flow rate are calculated by the value of sensor [34]. Based on [3,33,34,35], in normal conditions, the nonlinear feedback ratio controller is used to hold the air-to-fuel ratio to 14.6. In case of a fault in any sensor at a time, the look-up table is used for sensor’s value estimation. Based on [34], normal and abnormal data were extracted from ICE when the motor speed changed from 300 to 600 rpm. Figure 2 illustrates the system used for sensor data collection. Moreover, the air-to-fuel-ratio signal was used to test the condition of the sensors. In this work, 12 conditions were analyzed: normal condition (NC), throttle fault condition (TFC), speed fault condition (SFC), EGO fault condition (EFC), MAP fault condition (MFC), throttle-speed fault condition (TSFC), throttle-EGO fault condition (TEFC), throttle-MAP fault condition (TMFC), speed-EGO fault condition (SEFC), speed-MAP fault condition (SMFC), EGO-MAP fault condition (EMFC), and throttle-speed-EGO-MAP (ALL) fault condition (AFC). The original signal has 240,000 samples for 12 conditions when the sampling rate is 100 kHz. All simulations are performed in MATLAB 2015a software with the system configuration of Intel (R) core™ i7-7500U, 8 GB RAM, 2.7 GHz processor, and 64-bit Windows 10 operating system.

Table 1 illustrates these 12 conditions according to states of sensors in the ICE. In this table, when the sensor value is 1, the sensor is working normally. When the sensor is working abnormally, the corresponding value will be 0 [3,33,34,35]. Moreover, an ICE system has four single faults: TFC, SFC, EFC, and MFC. In addition, it has seven multi-faults: TSFC, TEFC, TMFC, SEFC, SMFC, EMFC, and AFC. Figure 3 shows the original air-to-fuel-ratio signals in all the above conditions.

In this figure, it is difficult to detect the single faults (TFC, SFC, EFC, and MFC). Furthermore, multi-fault detection (TSFC, TEFC, TMFC, SEFC, SMFC, EMFC, and AFC) is the other challenge with an ICE. Figure 4 shows the resampled RMS air-to-fuel-ratio signal under the 12 conditions.

Figure 5 illustrates the error in ICE sensor modeling using the GA approach, the GA-Laguerre scheme, and the fuzzy GAL method under normal conditions.

Based on Figure 5, the error with the proposed FGAL method under normal conditions was lower than the other two techniques. The fuzzy technique with rule-bases increases the strength, flexibility, and stability and reduces the error of the signal modeling compared with the other two approaches. Moreover, the robustness of ICE sensor modeling under normal conditions is shown in Figure 6. To test robustness, the ICE sensors modeled using the GA approach, the GAL scheme, and the fuzzy GAL method were tested when ICE speed changed between 300 rpm and 600 rpm.

Based on Figure 6, the fluctuations of the GAL scheme and the proposed FGAL method were lower than that of the GA approach. According to Figure 6, the combination of fuzzy approach, Gaussian AR technique, and Laguerre filter reduces the dependence of ICE modeling to the variation of the speed engine. This will increase the robustness, stability, and reliability. Based on Figure 5 and Figure 6, the proposed FGAL method had better performance (errors in modeling and robustness) than the other two approaches.

To test the effectiveness of the proposed FGAL-FLCR technique, it was compared with two state-of-art techniques including FGAL-CR technique and the FGAL-LCR technique and three existing methods. Figure 7, Figure 8 and Figure 9 show the resampled RMS residual signals for air-to-fuel ratio under the above 12 conditions using the FGAL-CR technique, the FGAL-LCR method, and the proposed FGAL-FLCR approach, respectively.

Based on this figure, this technique has issues with fault classifications between some states, such as NC and TFC plus TMFC and SEFC. Apart from the advantages of CR observer such as significant performance in certain conditions, this technique is suffering from robustness. To reduce overlapping and to improve the signal identification’s robust algorithm, we need a robust algorithm. Moreover, Figure 8 shows resampled RMS residual signals for air-to-fuel ratio using the FGAL-LCR method. This technique is more robust than the previous method, but it has a challenge in fault classification between states, too, especially between TSFC and TEFC.

The combination of the computed ratio observer and Lyapunov approach will increase the robustness and reliability, compared to the computed ratio technique. By comparing Figure 7 and Figure 8, it can be seen that FGAL-LCR performs a better signal identification. Figure 9 illustrates resampled RMS residual signals for air-to-fuel ratio using the proposed FGAL-FLCR technique. The discrimination power from the proposed method is much better than the other two techniques.

Based on Figure 7, Figure 8 and Figure 9, the discrimination powers from the FGAL-CR technique, the FGAL-LCR method, and the proposed FGAL-FLCR approach to resampled RMS residual signals for air-to-fuel ratios in the above 12 conditions are illustrated, and we can see that the proposed FGAL-FLCR has better performance. To test the accuracy of the resampled RMS residual signal for air-to-fuel ratio classification, the SVM technique was used in parallel with the FGAL-CR, FGAL-LCR, and FGAL-FLCR approaches. Figure 10, Figure 11 and Figure 12 illustrate the confusion matrices for FGAL-CR with SVM, FGAL-LCR with SVM, and FGAL-FLCR with SVM, respectively.

Based on Figure 10, the average accuracy of the resampled RMS residual signals for air-to-fuel ratio classification using FGAL-CR and the SVM was 89.8%. Based on this figure, the classification accuracy for SEFC was 85% with 15% misclassifications for TMFC and SMFC. Based on Figure 7 and Figure 10, it is clear that this technique is suffering from robustness.

According to Figure 11, the average accuracy of the resampled RMS residual signals for air-to-fuel ratio classification using FGAL-LCR and an SVM was 96%. It is clear that FGAL-LCR is more robust than FGAL-CR. By the combination of CR observer and Lyapunov approach, the signal identification accuracy can be increased.

To reduce the effect of uncertainties, the fuzzy logic approach was used for this step. The combination of the computed fuel ratio observer, Lyapunov approach, and fuzzy technique results in the increase in signal classification accuracy. The average accuracy of the resampled RMS residual signals for air-to-fuel ratio classification using the proposed FGAL-FLCR and SVM was 98.17%. Based on Figure 10, Figure 11 and Figure 12, the proposed FGAL-FLCR and SVM improved the average accuracy of the resampled RMS residual signals for sensor state classification by 8.37% and 2.17%, compared to FGAL-CR with SVM and FGAL-LCR with SVM, respectively. Moreover, to test the robustness of the proposed FGAL-FLCR, FGAL-CR, and FGAL-LCR methods, we processed the training and test data 20 times randomly and found the average accuracy from classification each time. Figure 13 shows the robustness tests of FGAL-CR, FGAL-LCR, and the proposed FGAL-FLCR.

Based on Figure 13, the proposed FGAL-FLCR approach is more robust than the other two methods. Thus, Figure 10, Figure 11, Figure 12 and Figure 13 show that the proposed FGAL-FLCR observer is more robust and accurate than the other two methods for classification of ICE sensor signals under different types of condition. All simulations were performed in MATLAB 2015a software with the system configuration of Intel (R) core™ i7-7500U, 8 GB RAM, 2.7 GHz processor and 64-bit Windows 10 operating system. Regarding Figure 1, the proposed fuzzy Lyapunov-based computed ratio observer has two main parts: signal modeling and signal estimation. For signal modeling, the Gaussian AR technique, Gaussian AR-Laguerre technique, and fuzzy Gaussian AR-Laguerre technique were compared. Regarding the results, the consumption times for the Gaussian AR (GA) technique, Gaussian AR-Laguerre (GAL) technique, and fuzzy Gaussian AR-Laguerre technique (FGAL) are 0.223 s, 0.224 s, and 0.47 s, respectively. For signal estimation, the FGAL-CR technique, FGAL-LCR approach, and the proposed FGAL-FLCR scheme are compared. As a result, the consumption time for the FGAL-CR technique, FGAL-LCR approach, and the proposed FGAL-FLCR scheme are 1.49 s, 1.49 s, and 1.8 s, respectively.

In the next part, to validate the effectiveness of the proposed method (FGAL-FLCR), the proposed approach is compared with the following four existing methods including feedback linearization algorithm [3], self-tuning network-fuzzy robust proportional multi-integral and smart autoregressive model technique [11], and Kalman filter with advanced redundancy [35]. In [3], the authors used the combination of fuzzy ARX-Laguerre modeling technique and fuzzy feedback linearization observation (FFLO) algorithm for single type fault diagnosis. The main challenge of this technique is robustness. In addition, in [11], the combination of smart autoregressive technique and self-tuning fuzzy robust multi-integral observation (SFMIO) technique was suggested for fault diagnosis. In this approach the authors utilized a linear-based observer. The main challenge of this technique is to perform multi-type fault diagnosis. Furthermore, the Kalman filter (KF) was used for fault detection and identification [35]. However, KF is a model-based observer, and it has a challenge for fault diagnosis of complex signal. To validate our proposed method, we calculate the average diagnostic accuracy for each sensor fault state (see Table 2) under various operating conditions. Table 2 shows the diagnostic accuracy of the FGAL-FLCR, FFLO [3], SFMIO [11], SCO [4], and KF [35] in various motor speeds. The diagnostic accuracy is measured as the percentage of the correct detection in all data.

As shown in Table 2, the proposed FGAL-FLCR fault diagnosis method outperforms the state-of-the-arts FFLO, SFMIO, and KF methods, while yielding average performance improvements of 6.17%, 4.57%, and 5.37% for different ICE speeds, respectively. This performance improvement can be further validated by the fact that our proposed FGAL-FLCR method is highly sufficient to identify the signal state and sensor fault conditions.

## 4. Conclusions

The principal goal of this research was to solve the challenge of sensor fault diagnosis in the ICE. A fuzzy Gaussian autoregressive-Laguerre approach and the proposed fuzzy Lyapunov-based computed ratio observer, combined with an SVM, were used to solve this problem. The proposed machine fuzzy Lyapunov-based computed ratio observer has three layers. The first layer (preprocessing) extracts the resampled RMS features from the original fuel-to-air-ratio signals. The second layer is the main part of the proposed sensor-fault-classification technique. This layer has two sub-layers: modeling and estimation. To model resampled RMS normal signals for air-to-fuel ratio in an ICE, the proposed robust Gaussian autoregressive technique was improved with the Laguerre filter and a fuzzy approach, while the subsequent sub-layer was used for resampled RMS fuel-to-air-ratio signal estimation using the proposed robust fuzzy Lyapunov-based computed ratio observer. This layer is formed to overcome the nonstationary and nonlinear behavior of the ICE sensor signals. In the third layer, the residual signals are determined, and SVM is used for sensor fault diagnosis. The residual signal generator was used to find the difference between the original resampled RMS signals and the estimated ones. It is observed that the proposed approach is able to estimate the normal and abnormal signals, which results in a superior diagnostic performance of the proposed model for fault pattern identification. ICE was modeled in MATLAB, and data containing the normal condition and 11 faulty conditions, with motor speeds varying between 300 and 600 rpm, were used to validate the proposed approach. Moreover, the proposed fuzzy Lyapunov-based computed ratio observer was compared with five fault identification approaches (i.e., a Lyapunov-based computed ratio observer, a computed ratio observer, fuzzy feedback linearization observation, self-tuning fuzzy robust multi-integral observer, and Kalman filter technique). The proposed method outperformed the Lyapunov-based computed ratio observer, a computed ratio observer, fuzzy feedback linearization observation, self-tuning fuzzy robust multi-integral observer, and Kalman filter technique, while yielding average performance improvements of 2.17%, 8.37%, 6.17%, 4.57%, and 5.37%, respectively. The results illustrated how the proposed fuzzy Lyapunov-based computed ratio observer is more effective, compared to the other five approaches, regardless of nonlinearity in the resampled RMS fuel-to-air-ratio signals due to various types of sensor faults. However, from the perspective of misclassification, the performances of the pro-posed algorithm for throttle-EGO and speed-MAP fault conditions were slightly reduced, which underscores the need for a more robust observation algorithm in the future. In the future, combining machine/deep learning and a slight smooth sliding approach will be introduced to improve stability, robustness, and accuracy from sensor anomaly diagnoses.

## Figures and Tables

**Figure 1 sensors-22-02974-f001:**
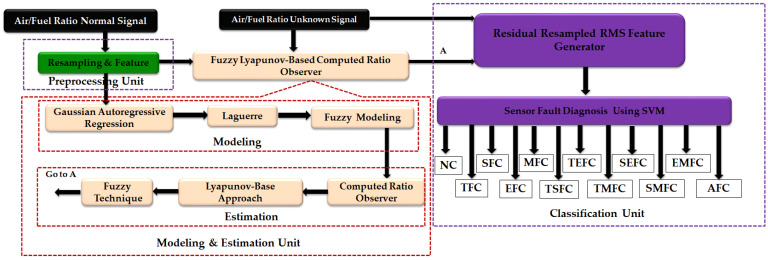
ICE sensor fault diagnosis using the machine fuzzy Lyapunov-based algorithm.

**Figure 2 sensors-22-02974-f002:**
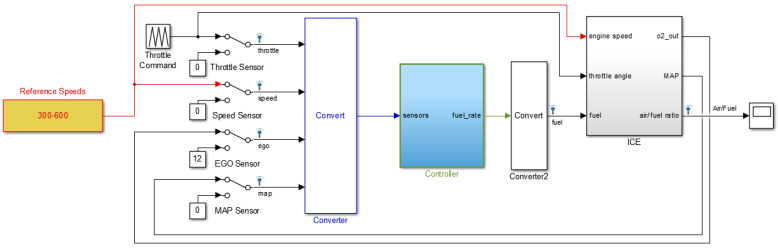
Collecting data from a system with sensors in normal and abnormal conditions.

**Figure 3 sensors-22-02974-f003:**
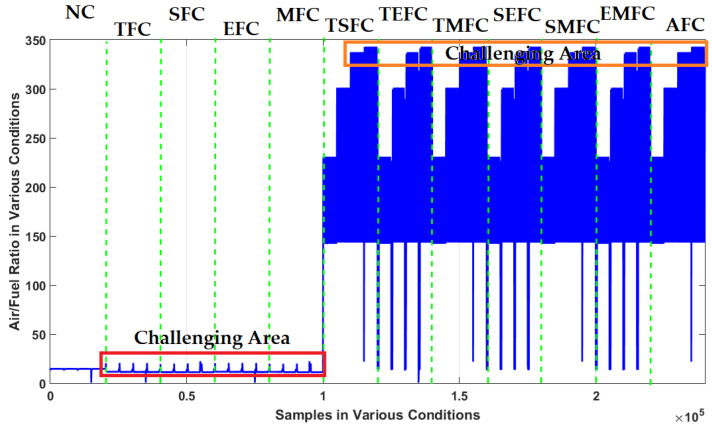
Air-to-fuel-ratio signal when sensors operate in normal and abnormal conditions.

**Figure 4 sensors-22-02974-f004:**
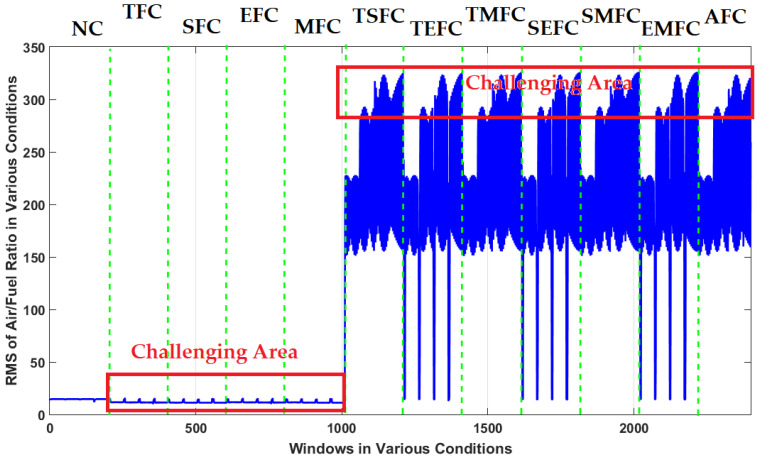
RMS resampled air-to-fuel-ratio signals when sensors operate under normal and abnormal conditions.

**Figure 5 sensors-22-02974-f005:**
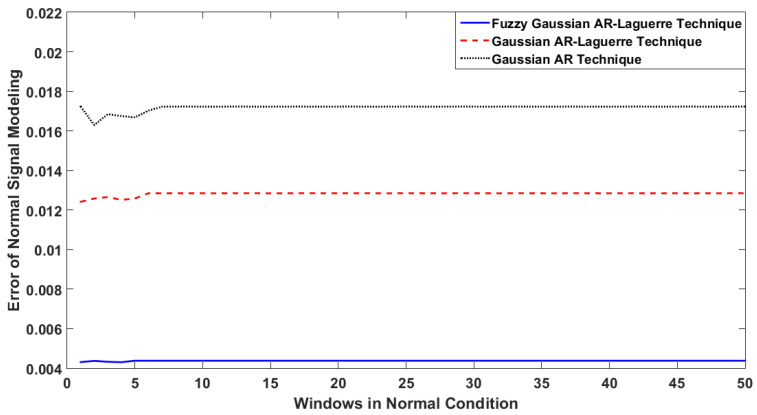
Error in ICE sensor modeling under normal condition according to the GA approach, the GA-Laguerre scheme, and the fuzzy GAL method.

**Figure 6 sensors-22-02974-f006:**
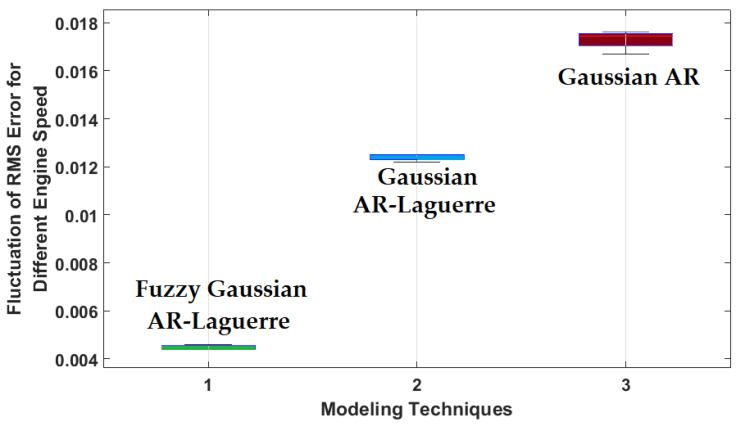
Fluctuations in ICE sensor modeling under normal conditions according to the GA approach, the GAL scheme, and the FGAL method.

**Figure 7 sensors-22-02974-f007:**
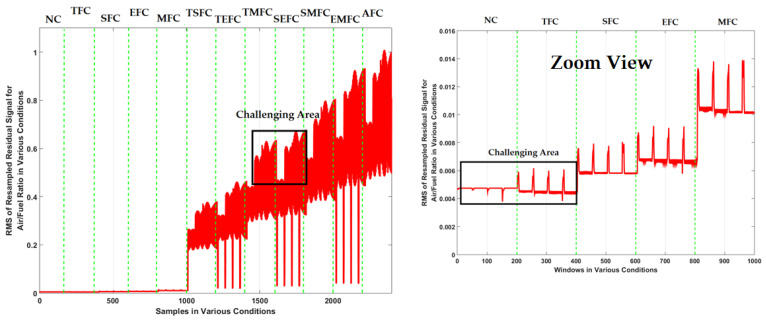
Resampled RMS residual signals for air-to-fuel ratio using the FGAL-CR technique.

**Figure 8 sensors-22-02974-f008:**
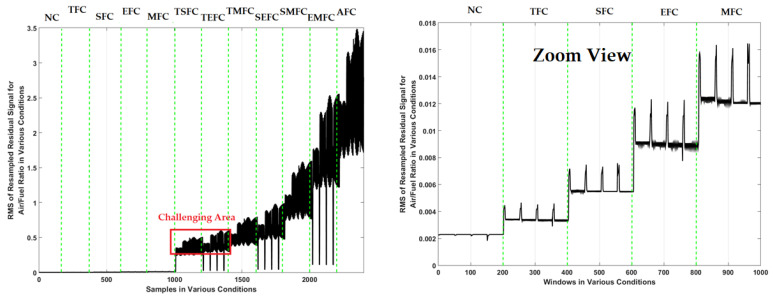
Resampled RMS residual signals for air-to-fuel ratio using the FGAL-LCR technique.

**Figure 9 sensors-22-02974-f009:**
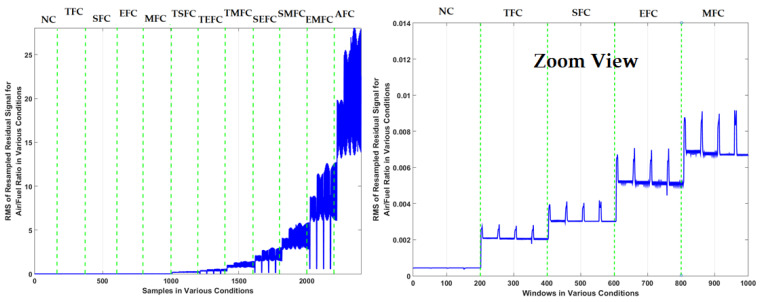
Resampled RMS residual signals for air-to-fuel ratio using the proposed FGAL-FLCR technique.

**Figure 10 sensors-22-02974-f010:**
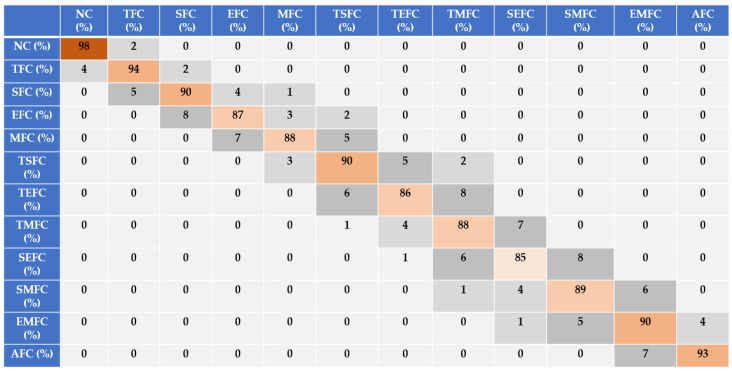
Sensor-fault-classification accuracy using FGAL-CR and an SVM.

**Figure 11 sensors-22-02974-f011:**
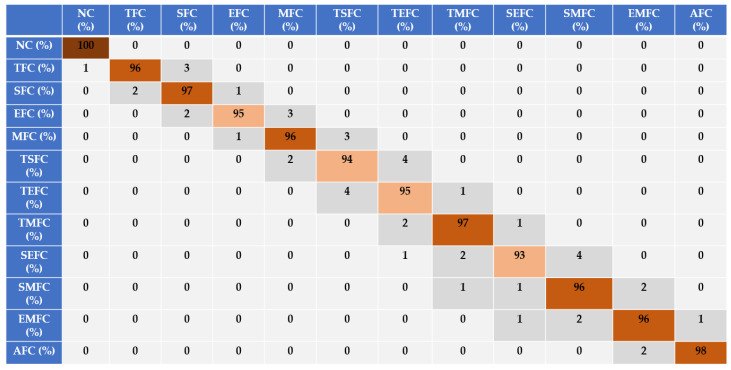
Sensor-fault-classification accuracy using FGAL-LCR and an SVM.

**Figure 12 sensors-22-02974-f012:**
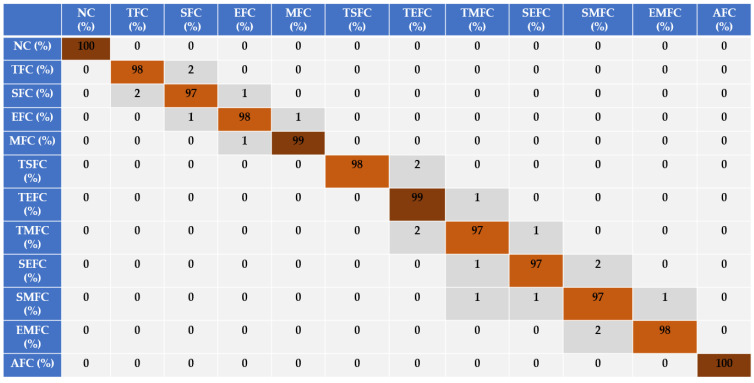
Sensor-fault-classification accuracy using the proposed FGAL-FLCR and an SVM.

**Figure 13 sensors-22-02974-f013:**
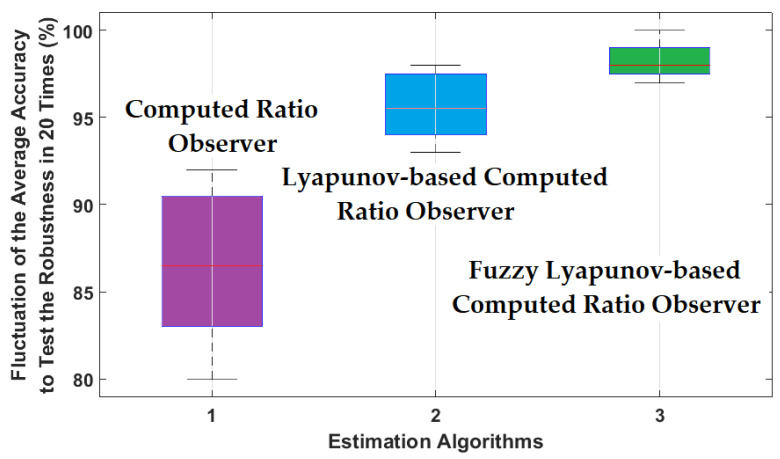
Fluctuation in average accuracy, which indicates the robustness of the FGAL-CR method, the FGAL-LCR technique, and the proposed FGAL-FLCR approach.

**Table 1 sensors-22-02974-t001:** States of sensors in the ICE.

Throttle Sensor	Speed Sensor	EGO Sensor	MAP Sensor	Motor Speed (RPM)	Condition
1	1	1	1	300–600	NC
0	1	1	1	300–600	TFC
1	0	1	1	300–600	SFC
1	1	0	1	300–600	EFC
1	1	1	0	300–600	MFC
0	0	1	1	300–600	TSFC
0	1	0	1	300–600	TEFC
0	1	1	0	300–600	TMFC
1	0	0	1	300–600	SEFC
1	0	1	0	300–600	SMFC
1	1	0	0	300–600	EMFC
0	0	0	0	300–600	AFC

**Table 2 sensors-22-02974-t002:** Average accuracies of sensor fault diagnosis according to the proposed FGAL-FLCR, FFLO, SFMIO, and KF approaches.

Conditions	FGAL-FLCR (%)	FFLO (%) [3]	SFMIO (%) [11]	KF (%) [35]
NC	100	100	100	100
TFC	98	95	93	92
SFC	97	90.7	93	91
EFC	98	89	94	94
MFC	99	91	94	92
TSFC	98	91	95	97
TEFC	99	90	92	95
TMFC	97	90	90	90
SEFC	97	93	91	91
SMFC	97	94	90	90
EMFC	98	91	93	90
AFC	100	90	98	92
Average	98.17	92	93.6	92.8

## Data Availability

The data are publicly available.

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
