# Peer review of "Sensor Fault Diagnosis Using a Machine Fuzzy Lyapunov-Based Computed Ratio Algorithm"

_sensors, 2022, doi:10.3390/s22082974_

Round 1

Reviewer 1 Report

The authors proposed an indirect fuzzy Lyapunov-based computed ratio observer integrated with an SVM for sensor fault
classification. In my point of view, the paper is well-written and could be published without changes.

The paper is well written and presents a good advance in the field. In order to improve the paper and be reproducible, I have a few comments for the authors.

1) Methods based on Machine learning are well-known to be a time-consuming task. The authors should provide a time comparison among the methods and specify the hardware/software descriptions used.

2) Based on the above, could this methodology be applied to real-time fault identification? Or how could you adapt this method to be applied in a real engine?

Author Response

Author's Reply to the Review Report (Reviewer 1)

The authors’ responses to reviewer comments are in blue bold.

Manuscript-ID: Sensor-1668169

Title: Sensor Fault Diagnosis Using a Machine Fuzzy Lyapunov-based Computed Ratio Algorithm

Authors: Shahnaz Tayebihaghighi and Insoo Koo

First, we would like to thank the editor and the reviewers for their precious time for reviewing our paper and providing us with their valuable comments. The comments were immensely helpful in improving the quality of our manuscript. We have revised the manuscript and have carefully addressed all the comments. The corresponding changes and refinements we made in the revised manuscript are summarized in our response below.

Reviewer 2 Report

The Manuscript ID sensors-1668169, titled “Sensor Fault Diagnosis Using a Machine Fuzzy Lyapunov-based Computed Ratio Algorithm” belongs to the field of Sensors anomaly detection techniques subdivision the accuracy of sensor anomaly classification.   

Concerning the scientific content of the paper

  1. The Introduction must be consistent, and it is, and the paragraph “2. The Proposed Scheme “ is also well detailed and emphasized.
  2. Unfortunately, the paragraph “ Data and Results” must be rewritten because is not contain any reference to the theoretical (simulated) data and experimental data in the literature.
  3. For the reader, it would be better to introduce a new paragraph, “Discussion”. In this new paragraph, the authors must comment on the obtained results using their models and techniques with the results obtained by other researchers in the field of sensors fault designed for ICE.
  4. In paragraph “ Conclusions”, the novelty brought by the authors to solve the challenge of sensor fault diagnosis for the ICE is not emphasized enough.
  5. Why did the authors choose only the range of rotation velocity of the ICE’s crankshaft 300-600 rpm ( this being only for industrial ICE), while the range for rotation velocity crankshaft of automotive ICE is 750-6000 rpm?

About the presentation of the paper concerning the Manuscript Type MDPI journal template

  1. The References don’t respect 100% the form demanded by the Manuscript Type MDPI journal template ( for the articles, the year must be bolded, not put into brackets).

For the aspects mentioned above, I strongly advise the major revision of the paper.

Author Response

Author's Reply to the Review Report (Reviewer 2)

The authors’ responses to reviewer comments are in blue bold.

Manuscript-ID: Sensor-1668169

Title: Sensor Fault Diagnosis Using a Machine Fuzzy Lyapunov-based Computed Ratio Algorithm

Authors: Shahnaz Tayebihaghighi and Insoo Koo

First, we would like to thank the editor and the reviewers for their precious time for reviewing our paper and providing us with their valuable comments. The comments were immensely helpful in improving the quality of our manuscript. We have revised the manuscript and have carefully addressed all the comments. The corresponding changes and refinements we made in the revised manuscript are summarized in our response below.

Reviewer 3 Report

Anomaly identification for internal combustion engine (ICE) sensors has become an important research area in recent years. In this work, a proposed indirect fuzzy Lyapunov-based computed ratio observer integrated with a support vector machine (SVM) is designed for sensor fault classification. The work seems to be carefully done and has been well reported with the exception of some minor issues detailed below.

  • The results were not at all compared with the research of other authors. It is difficult to evaluate the correctness of the experiments and the results without comparison. It is necessary to find as close as possible research oriented in terms of materials and parameters, because in the present form it is only the presentation of the results.
  • In the part of Data and Results, the authors described the phenomenon of experimental results more, and some analysis on the mechanism and reasons of its occurrence was neglected.  
  • The technical features, measuring ranges and accuracy of the measuring devices are not stated.
  • The authors needs to describe the main results in the conclusions.

Author Response

Author's Reply to the Review Report (Reviewer 3)

The authors’ responses to reviewer comments are in blue bold.

Manuscript-ID: Sensor-1668169

Title: Sensor Fault Diagnosis Using a Machine Fuzzy Lyapunov-based Computed Ratio Algorithm

Authors: Shahnaz Tayebihaghighi and Insoo Koo

First, we would like to thank the editor and the reviewers for their precious time for reviewing our paper and providing us with their valuable comments. The comments were immensely helpful in improving the quality of our manuscript. We have revised the manuscript and have carefully addressed all the comments. The corresponding changes and refinements we made in the revised manuscript are summarized in our response below.

Round 2

Reviewer 2 Report

The Manuscript ID sensors-1668169 revised version, titled “Sensor Fault Diagnosis Using a Machine Fuzzy Lyapunov-based Computed Ratio Algorithm,” belongs to the field of Sensors anomaly detection techniques subdivision the accuracy of sensor anomaly classification.  

Concerning the scientific content of the paper

All the comments and suggestions were corrected in the revised form of the manuscript by the authors.

About the presentation of the paper concerning the Manuscript Type MDPI journal template

The newly revised form of the manuscript respects the Manuscript Type of the MDPI journal template entirely.

For the aspects mentioned above, the manuscript has the content and the form to be published in SENSORS.